# Unveiling the Probiotic Potential of the Anaerobic Bacterium *Cetobacterium* sp. nov. C33 for Enhancing Nile Tilapia (*Oreochromis niloticus*) Cultures

**DOI:** 10.3390/microorganisms11122922

**Published:** 2023-12-05

**Authors:** Mario Andrés Colorado Gómez, Javier Fernando Melo-Bolívar, Ruth Yolanda Ruíz Pardo, Jorge Alberto Rodriguez, Luisa Marcela Villamil

**Affiliations:** 1Doctorado en Biociencias, Facultad de Ingeniería, Universidad de La Sabana, Chía 250001, Colombia; mario.colorado@shaio.org (M.A.C.G.); javiermebo@unisabana.edu.co (J.F.M.-B.); ruth.ruiz@unisabana.edu.co (R.Y.R.P.); jorge.rodriguez1@unisabana.edu.co (J.A.R.); 2Fundación Clínica Shaio, Bogotá 110121, Colombia

**Keywords:** probiotics, fish microbiome, *Cetobacterium*, freshwater fish, whole genome sequencing, Nile tilapia, anaerobic bacteria

## Abstract

The bacterium strain *Cetobacterium* sp. C33 was isolated from the intestinal microbial content of Nile tilapia (*O*. *niloticus*) under anaerobic conditions. Given that *Cetobacterium* species are recognized as primary constituents of the intestinal microbiota in cultured Nile tilapia by culture-independent techniques, the adaptability of the C33 strain to the host gastrointestinal conditions, its antibacterial activity against aquaculture bacterial and its antibiotic susceptibility were assessed. The genome of C33 was sequenced, assembled, annotated, and subjected to functional inference, particularly regarding pinpointed probiotic activities. Furthermore, phylogenomic comparative analyses were performed including closely reported strains/species relatives. Comparative genomics with closely related species disclosed that the isolate is not phylogenetically identical to other *Cetobacterium* species, displaying an approximately 5% sequence divergence from *C*. *somerae* and a 13% sequence divergence from *Cetobacterium ceti*. It can be distinguished from other species through physiological and biochemical criteria. Whole-genome annotation highlighted that *Cetobacterium* sp. nov. C33 possesses a set of genes that may contribute to antagonism against competing bacteria and has specific symbiotic adaptations in fish. Additional in vivo experiments should be carried out to verify favorable features, reinforcing its potential as a probiotic bacterium.

## 1. Introduction

The aquaculture industry is considered the fastest-growing industry in several countries worldwide and represents about 17% of global protein intake as reported by the FAO [1]. The increased demand for fish in the world market has led to production intensification with the increase in stocking densities being associated with stress factors and fish susceptibility to pathogens [2]. Therefore, in recent years strategies based on the manipulation of intestinal microbiota balance have been proposed to improve fish survival and growth [3]. Among these strategies, probiotics selection with mechanisms of action such as modulatory effect on the intestinal microbiota, the production of antimicrobial metabolites, vitamins, and enzymes, and immune regulation have been employed.

Numerous studies have highlighted the significance of the composition of intestinal bacteria for animal health due to its crucial role in protecting against infectious diseases and in maintaining the host’s immune and metabolic homeostasis [3,4,5]. Due to the complexity of the intestinal microbiota of humans and other animals, probiotics composed of more than one strain have mainly been developed with aerobic bacteria and yeast [6,7]. Even though most probiotic studies have been conducted to evaluate aerobic bacteria, it is known, mostly by culture-independent studies, that anaerobic bacteria are significant members of the fish intestinal microbiota.

Tsuchiya et al. [8] isolated vitamin B12-producing *Cetobacterium somerae* from the intestines of freshwater fish: goldfish, common carp, and Mozambique tilapia. Later, Ramírez et al. [9] reported *C*. *somerae* as a major component of the intestinal microbiota of the giant Amazonian freshwater fish, *Arapaima gigas*, using sequenced microbial DNA-based techniques.

Likewise, Melo-Bolivar et al. [10] reported the significant presence of Fusobacteria, represented mainly by *Cetobacterum* species, in *O*. *niloticus* microbial content from two different farms, and also in a continuous-flow competitive exclusion culture from the intestinal content of Nile tilapia juveniles using a metataxonomic DNA-based approach.

In addition, LaFrentz et al. [11] reported a *C*. *somerae* genome sequence from intestinal isolates of pond-raised channel catfish, *Ictalurus punctatus*. Recently, Xie et al. [12] evaluated the effects of stabilized *C*. *somerae* XMX-1 fermentation products on gut and liver health and zebrafish survival during a viral challenge; *C*. *somerae* XMX-1, was isolated from zebrafish intestines, cultured in anaerobic conditions, and added to the diet in a four-week feeding trial. It was found that dietary administration of *C*. *somerae* (XMX-1) improved the gut and liver health of zebrafish, reducing the intestinal inflammatory score, reducing proinflammatory cytokines, and increasing the antiviral gene expression; it also altered the composition of gut microbiota, reducing proteobacteria and increasing Firmicutes and Actinobacteria.

Zhang et al. [13] found that *Cetobacterium* was the core genus in the foregut, midgut, and hindgut of tilapia. They isolated *Cetobacterium* sp. NK01 from Nile tilapia foreguts, and sequenced the whole genome of the isolate, which indicated it to be a novel candidate species of the *Cetobacterium* sp. The genome analysis showed the production of amino acids, participating in various metabolic activities, and synthesizing vitamins, which indicated that *Cetobacterium* plays a key role in fish nutrition. However, the functions of *Cetobacterium* in the fish gut need to be further explored through in vivo and in vitro experiments [12,13].

In summary, probiotics are widely used in aquafeeds and exhibit beneficial effects in fish by improving host health and resistance to pathogens. Nevertheless, probiotics applied to aquaculture are mostly from terrestrial sources rather than the host animal and are mostly aerobic [14].

The purpose of the work was to isolate and characterize anaerobic bacteria from the gastrointestinal tract of Nile tilapia (*O*. *niloticus*) and to evaluate the probiotic potential in vitro.

## 2. Materials and Methods

### 2.1. Ethical Statement

The project followed the regulations of the Colombian national government. The Permit for access to genetic resources was issued by the Colombian Ministry of the Environment Number 117 (Otrosi N 4 RGE0154-4), on the 8 May 2018.

### 2.2. Bacteria Isolation

The *Cetabacterium* C33 strain was isolated from the intestinal microbiota of cultured Nile tilapia. Forty-seven samples were analyzed in which serial dilutions were conducted in a phosphate buffer (pH 7.3) containing 0.05% hydrochlorinated L-cysteine and 0.001% resazurin under anaerobic conditions [15]. Then, 100 μL was plated on Columbia agar at pH 7.22 with 5% lamb red blood cells, and incubated overnight under anaerobic conditions (O_2_: less than 1%; CO_2_: 9–13%; 28 °C) in an anaerobic jar (2.5 L AnaeroJar, Oxoid, Hampshire, UK). The colonies that showed different morphologies were sub-cultured in Columbia media following gram staining. Among the anaerobes isolated, the colonies that had a bacillus morphology and were gram-negative were selected. These selected samples were sub-cultured in Columbia media following the same method, and cryopreserved in 20% (*v*/*v*) glycerol under anaerobic conditions [16].

### 2.3. Phenotypic Characterization

The physiological and biochemical indices of the pure cultured C33 strain were evaluated using API 20A (BioMérieux, S.A., Marcy l’Etoile, France) according to the manufacturer’s instructions [13,17].

### 2.4. Whole-Genome Sequencing and Bacterial Identification

#### 2.4.1. DNA Extraction, Library Preparation, and Sequencing

Bacterial genomic DNA from the bacterial isolate C33 was extracted using the DNeasy^®^ UltraClean^®^ Microbial Kit (Qiagen, Hilden, Germany) following the manufacturer’s instructions. Briefly, bacteria were grown in Columbia Broth at 28 °C for 24 h. The DNA extraction method was optimized to reach a DNA concentration of 100 ng/DNA of C33 isolate. The sequencing library was prepared using the TruSeq Nano DNA kit. Finally, the paired reads were sequenced using the Illumina platform NovaSeq 6000.

#### 2.4.2. Quality Control, Trimming, Assembly of Paired Ends Reads, and Contigs Selection

The Paired-End Reads were assembled with Shovill v1.1.0 (SPAdes, v3.15.3; Velvet, v1.2.10; Megahit, v1.2.9; Skesa, v2.4.0) using default arguments (https://github.com/tseemann/shovill (accessed on 18 June 2023)). After obtaining the assembled contigs, a Quast comparison was used to select the assembly with the fewest contigs and an N50 length close to 50% of the total genome length [6,18].

#### 2.4.3. Bacterial Identification

JSpeciesWS v3.8.5 was used to identify the species (default parameters) through a Tetra correlation search along with ANIb (average nucleotide identity, calculated with the BLAST algorithm) and ANIm (average nucleotide identity, calculated with the alignment tool MUMmer) [19]. In addition to this approach, whole genome drafts were analyzed using the Type (Strain) Genome Server (TYGS) [20] to define species-level taxonomic affiliation.

RefSeq (NCBI Reference Sequence Database) was used to obtain the reference genomes of the bacteria. The assembled contigs were then loaded with their respective reference genomes into Medusa (http://combo.dbe.unifi.it/medusa (accessed on 20 July 2023)) [21] to determine the orientation and the order among the contigs to produce longer scaffolds [22,23]. The genome sequence data was uploaded to the Type Strain Genome Server (TYGS), a free bioinformatics platform available at https://tygs.dsmz.de (accessed on 26 July 2023), for full genome-based taxonomic analysis [24] and, for the determination of closely related type strains, a pairwise comparison of genomic sequences, phylogenetic inferences, and grouping of species and subspecies based on the type [24].

#### 2.4.4. Functional Annotation

Functional annotation of the genome and establishment of probiotic characteristics of the isolated and identified anaerobic bacteria was performed using programs such as the RAST (Rapid Annotation using Subsystem Technology) server for the identification of putative genes involved in tolerance to acid and bile salts, proteins potentially implicated in adhesion and aggregation, and genes important to intestinal survival, intestinal adhesion, and probiotic potential (https://rast.nmpdr.org/, accessed on 20 August 2023) [25]. The Resfinder program was also used for the identification of acquired resistance genes [26,27,28]. Regarding the detection of virulence genes, the Virulence Finder program was used [29]. The Mobile Element Finder tool enabled rapid detection of mobile genetic elements (MME) and their genetic context in assembled sequence data. MMEs are screened for sequence similarity against a database of 4452 known elements augmented with resistance gene annotation, virulence factors, and plasmid analysis [30]. Likewise, the online resource Virulence Factor Database (VFDB) for virulence factors [31] and antiSMASH were used for the rapid identification, annotation, and analysis of genes that biosynthesize secondary metabolites [32,33]. Finally, the BAGEL4 web server enabled the identification and visualization of gene clusters involved in the biosynthesis of ribosomal-synthesized post-translationally modified peptides (RiPP) and bacteriocins [34].

The raw reads used to assemble the draft genome were deposited in the sequence read archive (SRA) as PRJNA1010509. The genome sequence data was deposited under accession number JAVIKH000000000.

### 2.5. Evaluation of Probiotic Potential In Vitro

#### 2.5.1. Enzymatic Activity

To evaluate the enzymatic activity of C33 the Api-Zym galleries (BioMérieux, S.A., Marcy l’Etoile, France) were used [17,35].

#### 2.5.2. Vitamin B12 Production

The C33 strain was cultured in 100 mL of Columbia Broth medium under anaerobic conditions at 28 °C for 48 h, then the sample was filtered through 0.22 μm Millipore filters and transferred to an amber container at 4 °C; the sample was sent to the AOXLAB S.A.S laboratory, NIT 900.567.821-9 in Medellin Colombia; the Sample Code was Sample ID 9480-22. The analysis method used was AOAC 2011.09. Determination of vitamin B12 was evaluated using HPLC purification on an immunoaffinity column (1st Action) [36].

#### 2.5.3. Hemolytic Activity

The hemolytic activity was assessed following Melo-Bolívar et al. [6]. Briefly, the blood agar was prepared using Columbia agar (Condalab, Madrid, Spain) at pH 7.22, containing 0.05% hydro chlorinated L-cysteine and 0.001% resazurin under anaerobic conditions, with 5% (*v*/*v*) and sterile defibrinated sheep blood. The bacteria were seeded (100 μL at 1 × 10^8^ CFU/mL) onto the agar after the culture medium was solidified in the continuous flow of CO_2_. The Petri dishes were incubated at 28 °C for 48 h under anaerobic conditions (O_2_: below 1%; CO_2_: 9–13%) in an anaerobic jar (2.5 L AnaeroJar, Oxoid, Hampshire, UK).

#### 2.5.4. Bile Salts and pH Survival

This test was conducted following Melo-Bolívar et al. [6], with certain modifications. First, the Columbia broth (Condalab, Madrid, Spain) culture medium was prepared for pH resistance by adding 1 N HCl to a final pH of 2.0 or 3.0, containing 0.05% hydro chlorinated L-cysteine) and 0.001% resazurin under anaerobic conditions.

Bacterial survival in bile salts was evaluated in Columbia broth culture medium (Condalab, Madrid, Spain) adjusted to pH 7.0, containing 0.05% hydrochlorinated L-cysteine and 0.001% resazurin under anaerobic conditions, then a salt was added. Then, 0.3 percent *w*/*v* bile salts (Sigma-Aldrich, St. Louis, MO, USA) was added, and the medium was autoclaved. A saline solution (0.9 *w*/*v*) was used as a control. C33 isolate was then inoculated at 4.2 × 10^7^ CFU/mL in each treatment and incubated at 28 °C at 50 rpm. Agar plate counts were carried out every hour for three hours by inoculating 20 μL onto Columbia agar (Condalab, Madrid, Spain) containing 0.05% hydrochlorinated L-cysteine and 0.001% resazurin, and incubated in anaerobic conditions (O_2_: below 1%; CO_2_: 9–13%; 28 °C) in an anaerobic jar (2.5 L AnaeroJar, Oxoid, Hampshire, UK) at 28 °C for 36 h. The percentage of survival over time was estimated according to Equation (1) [6].
(1)$$\%\;Survival=\frac{bacterial\;concentration\;each\;treatment\;per\;hour\;(CFU/mL)\;}{bacterial\;concentration\;control\;at\;time\;0\;(CFU/mL)}\times 100$$

#### 2.5.5. Hydrophobicity Evaluation

The hydrophobicity of the isolates, as an indirect measure of adhesion ability, was determined using the Darilmaz et al., protocol [37]. Briefly, 2 mL of the bacteria (OD 0.08–0.10 at 600 nm in saline solution) (0.9 percent *w*/*v*) was vortexed for 1 min with 0.5 mL of chloroform in treatment one, or 0.5 mL ethyl acetate in treatment two. Then, the mixture was incubated for 10 min at 37 °C, the aqueous phase was removed, and the absorbance value was measured. The percentage of hydrophobicity was calculated using the following equation (2) [6]:(2)$$\%\;Hydrophobicity=\frac{\;{OD}_{600nm}\;before\;mixing-{OD}_{600nm}\;after\;mixing}{{OD}_{600nm}\;before\;mixing}\times 100\%$$

The hydrophobic activity of the evaluated strains was classified as high (51–100%), medium (30–50%), and low (0–29%), as proposed by Nader-Macías [38].

#### 2.5.6. Antibiotic Resistance

The minimum inhibitory concentration of antibiotics such as tetracycline, ampicillin, vancomycin, gentamicin, and chloramphenicol against the C33 bacterial strain was evaluated according to Florez et al. [39] and Melo-Bolívar et al. [6]. Colonies were suspended in sterile glass or plastic tubes containing 2 to 5 mL of sterile saline to a density corresponding to McFarland standard 1, or its spectrophotometric equivalent (approximately 10^8^ CFU/mL). A sterile cotton swab of the above McFarland suspension was spread on Columbia agar plates. After approximately 15 min, the E-test strips (BioMérieux, Durham, NC, USA) were applied. Following 48 h of incubation at 28 °C in anaerobic conditions, the results were classified as resistant (R) or susceptible (S) using the cut-off point recommended by the European Food Safety Authority (EFSA) according to the respective species and the inhibition zone [14].

#### 2.5.7. Antibacterial Activity against *Streptococcus agalactiae* and *Aeromonas hydrophila*

The antimicrobial activity of Extracellular Products (ECPs) obtained from C33 anaerobic bacteria was assessed against *S*. *agalactiae* and *A*. *hydrophila*, following the methodology previously described by Melo-Bolívar et al. [6]. The supernatants were centrifuged at 10,000× *g* for 30 min, and the ECPs were subsequently filtered, first with 0.45 µm Syringe Filters and then with 0.22 µm Syringe Filters to eliminate any remaining cells. The thermal stability of the ECPs was evaluated through two treatments: one involving heating at 80 °C for 3 min [40], and the other without a heating process. The dose–effect relationship was determined by diluting the ECPs to 50%, 25%, and 12%.

The experiments were conducted in 96-well plates, where pathogenic bacterial suspensions at a concentration of 6 × 10^8^ cells/mL (100 μL) were incubated with 100 μL of ECPs at different concentrations. Absorbance readings at 600 nm were taken every 60 min for a duration of 12 h, at 28 °C.

The survival rate was calculated as the absorbance percentage using Equation (3) [41]:(3)$$\;\;\;\;\;\;\;\;\;\;\;\;\;\;\;\;\;\;\;\%\;Inhibition=\frac{\left(Absorbance\;of\;control-Absorbance\;of\;test\right)}{Absorbance\;of\;control}\times 100$$

## 3. Results

### 3.1. Bacteria Characterization

Of the 47 samples of bacteria obtained from the tilapia intestinal content [10], sample C33 presented the typical characteristics of the *Cetobacterium* genus, therefore, it was selected for investigation due to the high presence reported in the intestinal microbiota in fish [8,9,10,11,12,13] and the beneficial properties that this genus has for Nile tilapia reported by several authors [12,13]. The C33 isolate was found to be a gram-negative bacillus, see Figure 1b, taken with a bright field microscope, and see Figure 1c,d, taken with a scanning electron microscope (SEM) (JEOL JSM-6460LV), which generates a round, white colony of 1.0 to 2.0 mm in diameter (Figure 1a) and grows in Columbia agar anaerobic medium as small colonies with an irregular shape, curved edge, umbonate elevation, moderate size, smooth texture, shiny appearance, and an opaque optical property.

Regarding the phenotypic characterization, the indole production test and gelatinase test for the C33 strain were negative, and the glucosidase test was positive. The urease reaction was negative and the fermentation of glucose, sucrose, maltose, salicin, mannitol, and trehalose was positive. However, the tests for mannitol, lactose, xylose, arabinose, glycerol, cellobiose, pine triose, raffinose, sorbitol, and rhamnose were negative, and the catalase reaction was negative (Table 1).

### 3.2. Whole-Genome Sequencing and Bacterial Identification

#### 3.2.1. Whole-Genome Sequencing

The counting assembly resulted in a scaffold level with a length of 2,830,091 bp (Figure 2a) and 2753 features were annotated: 2754 were protein-coding sequences and 40 were RNA genes, of which 22 were tRNA genes (Figure 2b). The G + C content of the C33 strain was 28.2 mol%.

#### 3.2.2. Bacterial Identification

When the whole-genome sequence of the C33 strain was compared to closely related strains, both the Average Nucleotide Identity, ANIb, and the digital DNA–DNA hybridization, dDDH, values were low (Table 2). The closest relative to the C33 strain in the analyses was *Cetobacterium somerae* ATCC BAA-474, with ANIb, ANIm, dDDH, and Tetranucleotide Signature Correlation Indices (Tetra) of 83.64, 86.21, 28.8%, and 0.9832, respectively. Similarly, in the analyses of *Cetobacterium* sp. NK01, also isolated from Nile tilapia, the ANIb, ANIm, dDDH, and Tetra were 83.97, 87,73, 28.9%, and 0.9822, respectively. Analysis of the C33 strain, using genome–genome comparisons with state-of-the-art approaches and up-to-date genomic and taxonomic reference databases such as the Genome Type (Strain) Server (TYGS), indicates that it belongs to the *Cetobacterium* genus, but its genome and 16S rRNA gene sequence phylogeny shows that it represents a new, undescribed species, having as its closest relatives both *C*. *somerae* and *C*. *ceti* (Figure 3). Therefore, we propose the name of Candidatus *Cetobacterium colombiensis* sp. nov. for the C33 strain.

#### 3.2.3. Functional Annotation as an In-Silico Tool for Probiotic Screening in Aquaculture

Moreover, the predicted genes were assigned to clusters of orthologous groups (COGs); these COG functional categories are compiled in Figure 2b. Based on the COG annotation results, the most gene-rich COG classifications were principally Carbohydrates (231), followed by Amino Acids and Derivatives (192), Protein Metabolism (159), Cell Wall and Capsule (141), Cofactors, Vitamins, Prosthetic Groups, Pigments (115), RNA Metabolism (102), Fatty Acids, Lipids, and Isoprenoids (61), Stress Response (58), Virulence, Disease and Defense (50) (Appendix A). The present article determines the possible presence of genes related to the following subsystems: amino acid and vitamin biosynthesis, adherent ability, carbohydrate utilization, and bacteriocin production. Amino acid biosynthesis of glutamine, glutamate, aspartate, asparagine, polyamine, methionine, threonine, homoserine, lysine, tryptophan, phenylalanine, tyrosine, proline, glycine, alanine, serine; and degradation of urea, histidine, arginine, ornithine, threonine, methionine, lysine, creatine, and creatinine was observed. Additionally, the vitamin biosynthesis of compounds like biotin, thiamin, cobalamin, heme, siroheme, riboflavin, flavodoxin, folate, and coenzyme A was also observed. Furthermore, Appendix A shows the possible presence of genes related to the metabolism of carbohydrates in the C33 bacteria strain.

The C33 strain featured the bacteriocin Linocin M18 and Zoocin A and two ribosomal sactipeptides (Table 3). C33 presented mobile genetic elements such as the tetA and tetB genes and Lnu(C) gene (Table 4). In addition, the C33 strain presented the plasmid named rep7a (Table 5).

### 3.3. Evaluation of Probiotic Potential In Vitro

#### 3.3.1. Enzymatic Activity

Enzyme activities were determined using an API-ZYM kit (BioMerieux) according to the manufacturer’s instructions. Positive enzymatic activities were determined for Esterase (C4) (2-naphthyl butyrate) and Acid Phosphatase (2-naphthyl phosphate) (Table 6).

#### 3.3.2. Vitamin B12 Production

The reported result indicates that the analyzed sample did not produce vitamin B12 as the value was below 0.05 µg/100 mL (quantification limit for this specific method).

#### 3.3.3. Hemolytic Activity

C33 is non-hemolytic; the hemolysis range (γ) was present.

#### 3.3.4. Bile Salts and pH Survival

The C33 strain survived in the presence of 0.3% bile salts (Figure 4). After three hours of incubation, a higher survival rate was observed in the treatment with pH 3.0 (48%) followed by pH 2.0 (39%) (Figure 5a,b).

#### 3.3.5. Hydrophobicity Evaluation

The hydrophobicity percentages of the C33 strain were higher in chloroform than when ethyl acetate was used, at 64% ± 1.15 and 8% ± 0.60, respectively.

#### 3.3.6. Antibiotic Resistance

The results of the Minimum Inhibitory Concentrations (MIC), expressed in μg/mL, obtained for the C33 strain are shown in Table 7. There is no report of a cut-off point for the *Cetobacterium* genus established by the European Food Safety Authority [14]. However, the breakpoints of a gram-negative bacillus (*Escherichia coli*) were used as a reference. In the context of probiotics, antibiotic susceptibility testing is performed to find the sensitivity and resistance of the probiotic strain against certain antibiotics that may be administered. The antibiotic susceptibility profile of the C33 isolate indicates that the strain was sensitive to all antibiotics evaluated in comparison to the cut-off value.

The C33 strain was more sensitive to Ampicillin, followed by Vancomycin, and Gentamicin, and less sensitive to Chloramphenicol (1.5 μg/mL) and Tetracycline (3.0 μg/mL) (Table 7).

#### 3.3.7. Antibacterial Activity against *S. agalactiae* and *A. hydrophila*

The pH of the extracellular products (ECPs) was at the value of 5.7. The ECPs of C33 showed antibacterial activity against *S*. *agalactiae* and *A*. *hydrophila*. In both pathogenic bacteria; a higher percentage of inhibition was observed in the ECPs that were not heated to 80 °C. However, no significant difference was observed between the different concentrations of ECPs. In contrast, the ECPs that were heated to 80 °C showed a significant difference between the different concentrations (Figure 6 and Figure 7).

## 4. Discussion

In the present study, we isolated the bacterial strain C33 from the intestinal content of Nile tilapia. It was previously found, in a continuous-flow competitive exclusion culture (CFCEC) obtained from Nile tilapia intestinal content, that Fusobacteria, primarily represented by the *Cetobacterum* genus, were highly abundant in the first days of the CFCEC [10].

C33 is an anaerobic gram-negative bacterium, that, according to the genome annotation, was identified as *Cetobacterium* sp. nov C33. Since it was found to be a dominant species, it in vitro analyses that could determine its potential use as probiotic were conducted.

The results of the phenotypic characterization of C33 showed gelatinase negative activity, which is favorable since it has been suggested that the presence of this activity in probiotic microorganisms could be detrimental to the health of the host due to the possible damage it can cause in the extracellular protein matrices of intestinal tissue [42]. In addition, the positive activity of esterase is related to the breaking activity of the ester bonds of polysaccharides, favoring the action of hydrolases of high molecular weight compounds such as carbohydrates and proteins. Likewise, the positive activity of acid phosphatase is important for the degradation of organic phosphorus found primarily in plant and animal protein sources [43]. In addition to these characteristics, the C33 strain has the possibility of using different energy sources such as glucose, sucrose, maltose, salicin, mannitol, and trehalose, a fact that may favor the fish’s nutrition, improving the absorption of nutrients by solubilizing the elements of the diets through extracellular enzymes; this facilitates the absorption of individual molecules through the intestinal epithelium of the animal host [44,45] and provides enzymes that the animal host does not have [44,46]. Furthermore, we have identified the C33 strain as a gamma (γ) hemolytic bacterium. In addition, for a bacterium to qualify as a probiotic, it must be able to survive and ideally colonize the intestine. This involves overcoming the stress generated by the low pH of the stomach and contact with bile salts, which are inhibitory for multiple microorganisms because they cause cell lysis [47,48,49,50,51]. Here we report the significant capacity of C33 to survive even at pH 2.0.

On the other hand, after passing through the digestive tract of the host, for a strain candidate to be used as a probiotic it must be able to colonize the gut. The C33 strain showed hydrophobicity percentages higher than 50% with a non-polar solvent (chloroform), a result that indicates a hydrophobic character of the cell surface, a factor that contributes to the interaction of the bacteria with the cells of the gastrointestinal tract. This indicates the potential ability of this isolate to colonize and persist in the gastrointestinal tract [6,41].

Antibiotic resistance and a growing reluctance to administer antibiotics have led to an increase in the use of probiotics [6,52]. Bacterial antibiotic resistance mechanisms can be innate, natural, or acquired. There is no horizontal transferability of the intrinsic resistance; nevertheless, the acquired resistance can be gained by mutations or the acquisition of genes through mobile genetic elements into their genomes [6,53,54]. The resistance to antimicrobial agents in a potential bacterial probiotic should be considered with caution. On the other hand, some authors mention that resistance to given antibiotics may be acceptable because, in the event of the use of any of these compounds being required in the fish culture, the probiotic will not be eliminated [6,55].

Nevertheless, the cutoff value of all the antibiotics tested in C33 indicated that this bacterium is sensitive. Additionally, in the RASfinder genome annotation, no resistance genes were found for any antibiotics (amoxicillin, ertapenem, aztreonam, amoxicillin, ampicillin, piperacillin, cefixime, ceftriaxone, ticarcillin, penicillin, cefotaxime, temocillin, metronidazole, doxycycline, minocycline, tigecycline, teicoplanin, and vancomycin). Due to the potential migration of antibiotic resistance factors, we consider that it is better to use a probiotic free of any acquired antibiotic resistance to avoid further changes in the microbial community induced by the overuse of the probiotic bacteria.

C33 presents with the plasmid rep7a, belonging to the group of glutathione S-transferases (GST), which are dimeric proteins that can conjugate glutathione (GSH) with a variety of compounds containing electrophilic centers. On the other hand, C33 also presented with the rep7a gene, the tetA and tetB genes related with tetracycline resistance, and the Lnu(C) gene that confers resistance to lincomycin [56].

C33 has two ribosomal sactipeptides (peptides cross-linked from sulfur to carbon alpha thioether) belonging to RiPP, that show various biological activities such as antibacterial properties and post-translationally modified peptides (RiPPs) [57]. Nevertheless, this condition requires further research for better understanding of its bacterial physiology.

The C33 strain has a gene encoding the Linocin M18 bacteriocin; a protein that forms nano compartments within the pathogenic bacterium, it also contains ferritin-like proteins or peroxidases and enzymes involved in the oxidative stress response. Various authors have reported that Linocin M18 has bacteriostatic activity against strains of *Arthrobacter*, *Bacillus*, *Brevibacterium*, *Corynebacterium*, and *Listeria* [58]. In addition, C33 presents with genes encoding Zoocin A, a peptidoglycan hydrolase, which, combined with lauricidin, a cell membrane active lipid, has been reported to selectively suppress the growth of *Streptococcus mutans* [59].

Here cell-free extracellular products (ECPs) of *Cetobacterium* sp. C33 are found to have antimicrobial activity against *S*. *agalactiae* and *A*. *hydrophila,* two pathogenic bacteria responsible for high mortality in Nile tilapia cultures and substantial economic losses in tilapia farms [48,60,61,62].

Probiotic use has been reported as an alternative to the use of antibiotics which has caused antimicrobial resistance in the aquaculture industry [63]. They contribute to natural resistance and a higher survival rate of the fish [64,65].

C33 has the capacity to biosynthesize amino acids, such as glutamine, glutamate, aspartate, asparagine, polyamine, methionine, threonine, homoserine, lysine, tryptophan, phenylalanine, tyrosine, proline, glycine, alanine, and serine, that benefit fish growth [66]. In addition, biotin production is important for the synthesis of fatty acids, the oxidation of carbohydrates, and the synthesis of purines. Biotin deficiency causes loss of appetite, dark coloration, and seizures in fish. Thiamine acts in Nile tilapia as a coenzyme in various metabolic decarboxylation reactions (pyruvic acid, alpha-ketoglutaric acid). Thiamine deficiency in fish results in weakness, terminal convulsions, degeneration, and fish fin paralysis. Tetrapyrrole compounds like heme, cobalamin, etc., have multiple essential functions in fish. Its synthesis begins with the formation of aminolaevulinic acid from glutamine [67]. Riboflavin and FAD are components of two enzymes (FMN and FAD) that are oxidases and reductases that participate in the metabolic degradation of proteins, carbohydrates, and lipids. Its deficiency in tilapia generates loss of appetite, dark skin, cataracts, and photophobia. Deficiencies in B vitamins, such as folate and vitamin B12, affect the offspring’s resistance and fertility [68].

In general, the C33 strain breaks down and uses Mannose, Chitin, N-acetylglucosamine, Sucrose, Maltose, Maltodextrin, Lactose, Galactose, Lactate, Glycerol, Glycerol-3-phosphate, D-ribose utilization, L-ascorbate, and Fructose. The genus *Cetobacterium* provides extracellular enzymes to degrade complex carbohydrates [69]. The presence of *Cetobacterium* may be associated with better glucose utilization [70] and could improve glucose homeostasis and increase insulin expression [71]. The above supports the role of *Cetobacterium* in carbohydrate regulation.

Finally, by comparing the sequencing results of the whole genome with the NCBI database, it was found that the C33 strain can be classified within the genus *Cetobacterium*. Similarly, the phylogenetic analysis indicates that the C33 strain has the greatest similarity with the species *Cetobacterium somerae* ATCC BAA-474, with a tetranucleotide signature correlation index (Tetra) result of 0.98712, a value that was less than 0.99 (Minimum value to consider it to be the same species [72]), which suggests that it can be considered a new species. In addition, the C33 strain differs from *C*. *somerae* WAL 14325T *Cetobacterium* sp. NK01 and *C*. *ceti* M 3333T in other aspects, such as no indole production, positive glucosidase reaction, positive enzyme activities for esterase (C4) (2-naphthyl butyrate) and acid phosphatase (2-naphthyl phosphate), and finally, it did not present any vitamin B12 production.

Consequently, the physiological characteristics and the phylogenetic analysis suggest that the C33 strain represents a new species of the genus *Cetobacterium* with high probiotic potential for Nile tilapia cultures based on the in vitro analysis shown here. Nevertheless in vivo experiments should be conducted in other studies to verify the effects on immune regulation, microbiota modulation fish growth, and resistance.

## 5. Conclusions

The anaerobic bacterial strain C33, isolated from the intestinal microbial content of Nile tilapia was sequenced, showing that it was a new candidate species of the genus *Cetobacterium* that could be named *Cetabacterium colombiensis* sp. nov C33.

Isolated C33 has probiotic characteristics which are high adaptability to gastrointestinal conditions, and a potential capacity to adhere to epithelial intestinal cells and produce antimicrobial substances.

To continue with the development of the probiotic product, the next step is to incorporate these bacteria into the fish feed and evaluate the effect on growth performance, microbiota modulation, and immunomodulation.

## Figures and Tables

**Figure 1 microorganisms-11-02922-f001:**
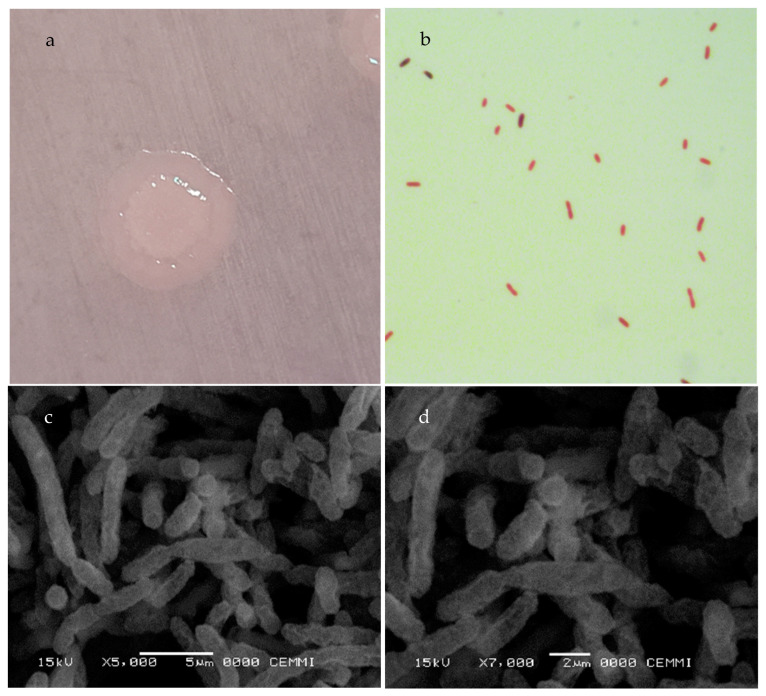
*Cetobacterium* sp. nov C33. (**a**) Colony morphology, 24 h culture in Columbia agar; (**b**) Gram staining, bright field microscopy (100×); SEM images of morphology and arrangement of C33 after 24 h culture in Columbia broth. (**c**) 5000× magnification and (**d**) 7000× magnification.

**Figure 2 microorganisms-11-02922-f002:**
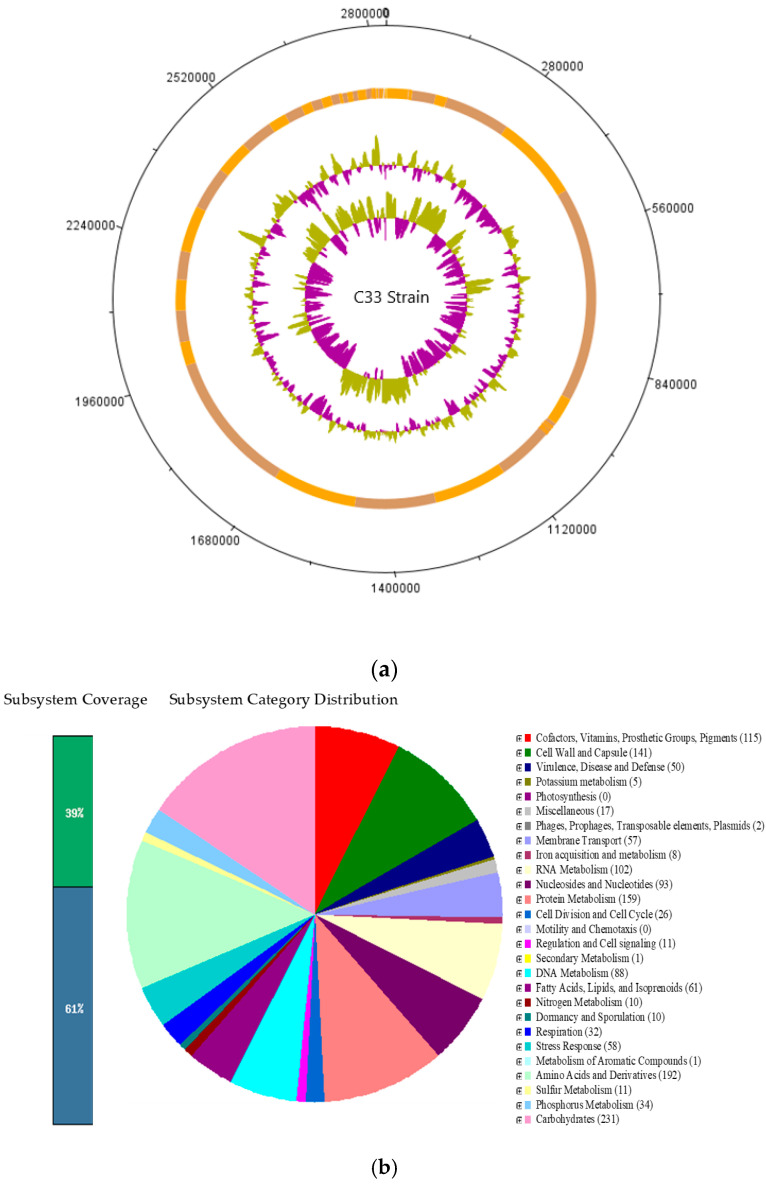
*Cetobacterium* sp. nov C33 whole-genome analysis. (**a**) Genomic atlas. Circles illustrate the following, from outermost to innermost rings: the scaffolds; the location of the contigs; the local % GC plot, and the innermost ring represents the GC skew. (**b**) Subsystem distribution of *Cetobacterium* sp. nov C33 based on Rapid Annotation using Subsystem Technology (RAST). Numbers mean: Number of Coding Sequences.

**Figure 3 microorganisms-11-02922-f003:**
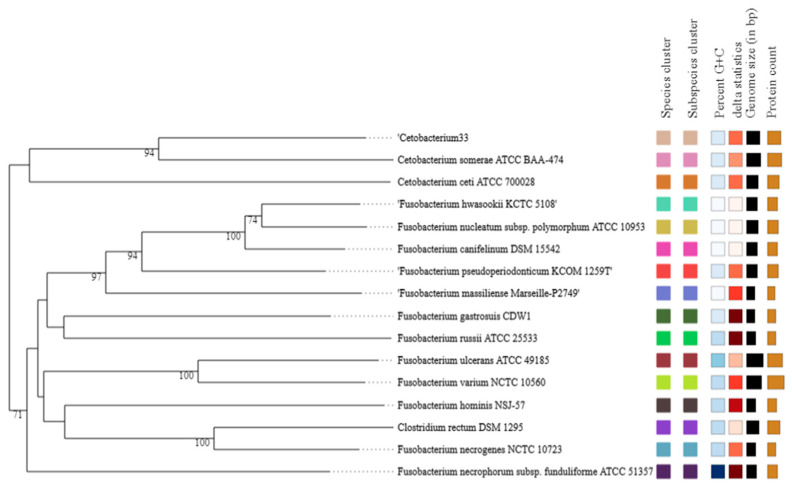
*Cetobacterium* sp. nov. C33 whole-genome sequence-based phylogenomic tree. The tree was inferred using FastME 2.1.6.1 with Genome Blast Distance Phylogeny approach (GBDP) distances calculated from genome sequences. The branch lengths were scaled in terms of GBDP-distance formula d5. The numbers below the branches are GBDP pseudo-bootstrap support values > 60% from 100 replications, with an average branch support of 73.1%. For the Percent G + C and Delta statistics columns, Darker color means higher value.

**Figure 4 microorganisms-11-02922-f004:**
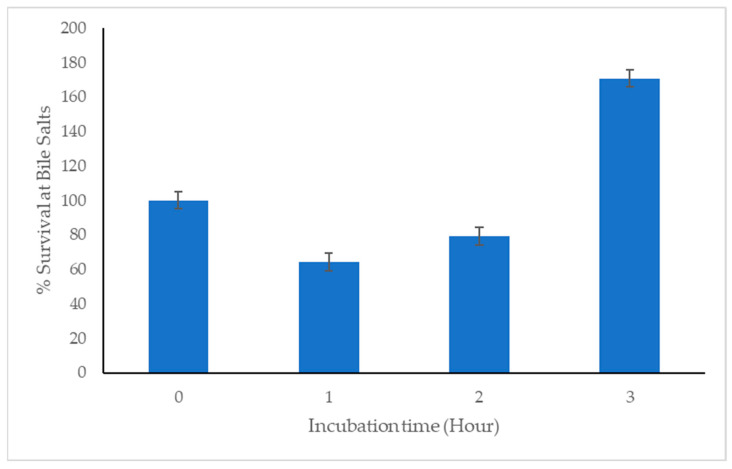
Evaluation of *Cetobacterium* sp. nov. C33 survival at 0.3 percent *w*/*v*% bile salts). Data represents mean + SEM (n = 3).

**Figure 5 microorganisms-11-02922-f005:**
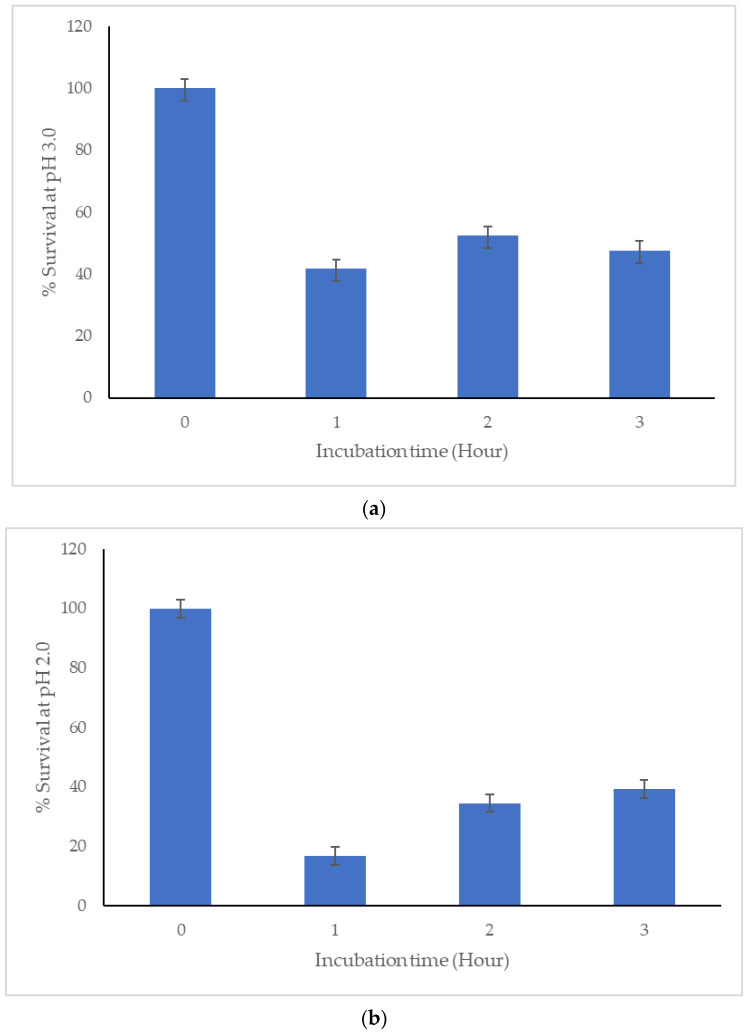
Evaluation of *Cetobacterium* sp. nov. C33 survival at (**a**) pH 3.0; (**b**) pH 2.0. Data represents mean + SEM (n = 3).

**Figure 6 microorganisms-11-02922-f006:**
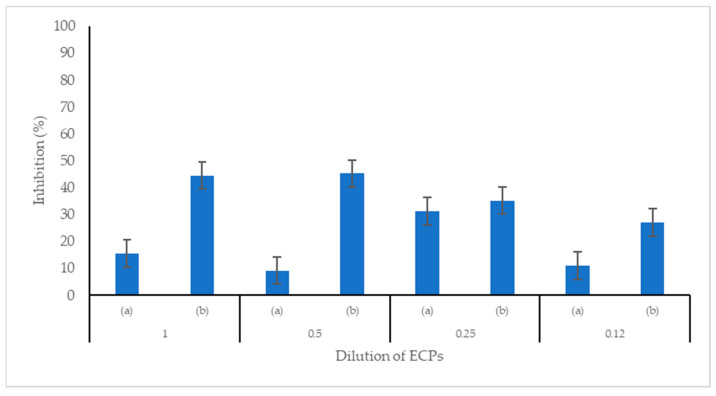
Antibacterial activity of the extracellular products (ECPs) of *Cetobacterium* sp. nov C33 against *Streptococcus agalactiae*: (a) ECPs heated at 80 °C; (b) ECPs without heating at 80 °C. Data represents mean + SEM (n = 3).

**Figure 7 microorganisms-11-02922-f007:**
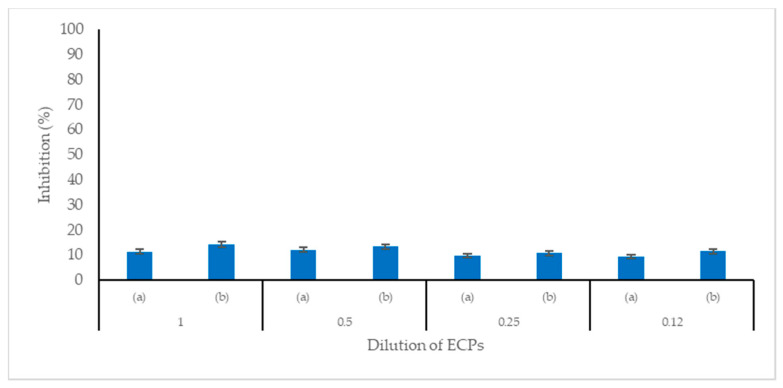
Antibacterial activity of the extracellular products (ECPs) of *Cetobacterium* sp. nov C33 against *Aeromonas hydrophila*: (a) ECPs heated at 80 °C; (b) ECPs without heating at 80 °C. Data represents mean + SEM (n = 3).

**Table 1 microorganisms-11-02922-t001:** *Cetobacterium* sp. nov. C33 phenotypic properties. (−) negative reaction, and (+) positive reaction.

Properties	*Cetobacterium* sp. nov. C33
Indole formation	−
Urease	−
Acidification (GLUcose)	+
Acidification (MANnitol)	−
Acidification (LACtose)	−
Acidification (SACcharose)	+
Acidification (MALtose)	+
Acidification (SALicin)	+
Acidification (XYLose)	−
Acidification (ARAbinose)	−
Hydrolysis (protease) (GELatin)	−
Hydrolysis (ß-glucosidase) (ESCulin)	+
Acidification (GLYcerol)	−
Acidification (CELlobiose)	+
Acidification (ManNosE)	+
Acidification (MeLeZitose)	+
Acidification (RAFfinose)	−
Acidification (SORbitol)	−
Acidification (RHAmnose)	−
Acidification (TREhalose)	−
Catalase	−
Vitamin B12	−
Spores	−
Gram reaction	−
Morphology	Rod

**Table 2 microorganisms-11-02922-t002:** *Cetobacterium* sp. nov. C33 pairwise genome comparisons vs. type strain genomes. Average Nucleotide Identity based on BLAST (ANIb); Average Nucleotide Identity values based on MUMmer algorithm (ANIm); digital DNA-DNA hybridization (dDDH).

Subject Strain	NCBI RefSeq	ANIb	ANIm	dDDH (d4, in %)	Tetra
*Cetobacterium somerae* ATCC BAA-474	GCA_000479045.1	83.64	86.21	28.8	0.98320
*Cetobacterium* sp. NK01	NCBI:txid2993530	83.97	87.73	28.9	0.98225
*Cetobacterium ceti* ATCC 700028	GCA_900167275.1	73.33	83.63	19.2	0.93057

**Table 3 microorganisms-11-02922-t003:** Gene clusters involved in biosynthesis in *Cetobacterium* sp. nov. C33 obtained from BAGEL4 web server.

Areas Of Interest (AOI)	Contig Position	Class
NODE_14_length_68805_cov_11741857.1.AOI_01	9860–29,860	Sactipeptides
NODE_14_length_68805_cov_11741857.1.AOI_02	21,395–41,395	Sactipeptides
NODE_12_length_73494_cov_14215396.25.AOI_01	24,074–44,827	3.3; Bacteriocin
NODE_13_length_71663_cov_18402088.3.AOI_01	38,945–59,284	93.3; Zoocin_A

**Table 4 microorganisms-11-02922-t004:** *Cetobacterium* sp. nov. C33 genetic elements obtained from Mobile Element Finder tool.

Gen	Phenotype	Accession	Counting	Contig Position	Coverage	Identity
tetB(P)	doxycycline, tetracycline, minocycline	NC_010937	NODE_39_length_5959_cov_111.589261_pilon	1788–3746	100.00%	97.54%
tetA(P)	doxycycline, tetracycline	AB054980	NODE_39_length_5959_cov_111.589261_pilon	1804–542	99.92%	92.72%
Inu(C)	Lincomycin	AY928180	NODE_61_length_1005_cov_102.149888_pilon	429–923	100.00%	98.99%

**Table 5 microorganisms-11-02922-t005:** *Cetobacterium* sp. nov. C33 plasmid obtained from Mobile Element Finder tool.

Name of Plasmid	Database	Accession	Counting	Position in Contig	Coverage	Identity
rep7a	Gram-positive	SAU83488	NODE_53_length_2280_cov_1.059013_pilon	1408–1976	95.31%	90.53%

**Table 6 microorganisms-11-02922-t006:** *Cetobacterium* sp. nov. C33 enzymatic activity analysis using Api Zym. (+) Positive result, (−) Negative result.

Enzyme Analyzed	*Cetobacterium* sp. nov. C33
Phosphatase alkaline	−
Esterase (C 4)	+
Esterase Lipase (C 8)	−
Lipase (C 14)	−
Leucine arylamidase	−
Valine arylamidase	−
Cystine arylamidase	−
Trypsine	−
α-chymotrypsine	−
Phosphatase acide	+
Naphtol-AS-BI-phosphohydrolase	−
α-galactosidase	−
ß-galactosidase	−
ß-glucuronidase	−
α-glucosidase	−
ß-glucosidase	−
N-acetyl-ß-glucosaminidase	−
α-mannosidase	−
α-fucosidase	−

**Table 7 microorganisms-11-02922-t007:** Antibiotic Minimum Inhibitory Concentrations (MIC) were obtained in *Cetobacterium* sp. nov. C33.

Antibiotic	MIC C33 Strain	* *Escherichia coli*
	(μg/mL)	(μg/mL)
Tetracycline	3 (S)	8
Ampicillin	0.016 (S)	8
Vancomycin	0.250 (S)	not reported
Gentamicin	0.750 (S)	2
Chloramphenicol	1.5 (S)	16

* Cut off values by EFSA (µg/mL) for *Escherichia coli*. Resistant (R) MIC > cut-off value; Susceptible (S) MIC < cut-off value.

## Data Availability

The raw reads used to assemble the draft genome were deposited in the Sequence Read Archive (SRA) under bio project accession number PRJNA1010509. Genome sequence data for C33 was deposited under accession number JAVIKH000000000.

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
