# Peer review of "Unveiling the Probiotic Potential of the Anaerobic Bacterium Cetobacterium sp. nov. C33 for Enhancing Nile Tilapia (Oreochromis niloticus) Cultures"

_microorganisms, 2023, doi:10.3390/microorganisms11122922_

Round 1

Reviewer 1 Report

Comments and Suggestions for Authors

This manuscript aimed to unveil the probiotic potential of an anaerobic bacterium. However, the manuscript was poorly written. The authors should revise the manuscript thoroughly. The comments are listed as follows;

1. Why was the pH survival evaluated at pH 2.0 or 3.0?

2. Why was the antibacterial activity evaluated toward Streptococcus agalactiae and Aeromonas hydrophila?

3. The sentence in Lines 169-170 was wrongly structured. Moreover, why was the culture heated at 80°C? Are you sure the centrifugation proceeded at 500 g?

4. The authors should provide information on how many strains were isolated and why strain C33 was selected.

5. The pictures of colony morphology, gram staining and bacterial morphology should be provided.

6. The figures should be revised to meet academic standards.

7. What was the letter in the last column in Table 3? Besides, the annotation of this table was wrong.

8. I suggest the authors evaluate the antibacterial activity with different concentrations of ECP.

9. Bacterial identification should be sectioned at the beginning of the Results.

10. Minor problems:

(1)   Delete the period in Line 96.

(2)   The unit of temperature should be checked thoroughly.

(3)   The bacteria amount in Line 116 should be checked.

(4)   Numbers should not appear at the beginning of a sentence.

(5)   The bacteria concentration in Lines 130-131 was wrongly written.

(6)   The superscript and downscript should be checked thoroughly.

(7)   The unit format in the manuscript should be uniform. Please check thoroughly.

(8)   The expression of equations should be revised.

(9)   The results were confused in Line 178.

(10)           The full name of all abbreviations should be mentioned when they appear for the first time in the Abstract or the Main text.

Comments on the Quality of English Language

Moderate editing of English language required

Reviewer 2 Report

Comments and Suggestions for Authors

The manuscript describes a previously undescribed anaerobic bacterial strain isolated from the digestive contents of Nile tilapia fish.
In my opinion, the present work, based on a wide range of analyses, for the most part, fulfills the requirements for publication.
My comments:
1. Figures 3 and 4 must be edited correctly as they are not well readable in their current form. In my PDF version, they are incorrectly positioned, with the text offset from the figures.
2. Lines 351-358 - this text should be in the Discussion. The sentence at lines 351-353 is repeated in the Discussion at lines 454-456.
3.  Lines 76 and 104 - incorrect citation order.

Round 2

Reviewer 1 Report

Comments and Suggestions for Authors

The authors have revised the manuscript according to the reviewers’ comments, and the quality is improved. However, I do not think my comments were addressed appropriately. The authors are not careful enough, and careless mistakes are easily found. This manuscript should be revised again more carefully to meet the publication standard of Microorganisms. I propose all the problems as follows just based on my previous comments in the first review round.

1. The authors cited reference 43 to prove that the pH in the stomach of the fish was between 2-3. However, the reference did not give such information. Besides, what is included in lines 342-344?

2. Almost all the revised contents were marked in the wrong lines. The authors should be careful.

3. The authors ignored the question of why the culture was heated at 80°C. Besides, I raised the doubt that the centrifugation proceeded at 500 g. Then, the authors changed it to 10,000 g, and many details of the method were changed. I am wondering if this is the method you used in your experiments. Why can you change it so casually?

4. The explanation for selecting strain C33 is not convincing.

5. Firstly, the colony morphology should be re-provided. Besides, Figures 1a, 1b, 1c and 2d should be put together.

6. I have told the authors that Table 6 should be revised. However, too many mistakes are still present: (1) the title of Table 6 should be revised; (2) the title of the second column should be MICs; (3) ug should be changed to mg; (4) the annotation ‘Resistant (R) MIC > CUT-OFF VALUE’ should be deleted as it did not appear in Table 6.

Comments on the Quality of English Language

Minor editing of English language required.

Author Response

Please see attached responses.
